# Analysis of Ocean Bottom Pressure Anomalies and Seismic Activities in the MedRidge Zone

Hakan S. Kutoglu [1] and Kazimierz Becek [2,*]

1 Department of Geomatics Engineering, Zonguldak Bulent Ecevit University, Zonguldak 67100, Turkey; shakan.kutoglu@beun.edu.tr
2 Faculty of Geoengineering, Mining and Geology, Wroclaw University of Science and Technology, 50-357 Wroclaw, Poland
* Correspondence: kazimierz.becek@pwr.edu.pl; Tel.: +48-884670998

**Abstract:** The Mediterranean Ridge accretionary complex (MAC) is a product of the convergence of Africa–Europe–Aegean plates. As a result, the region exhibits a continuous mass change (horizontal/vertical movements) that generates earthquakes. Over the last 50 years, approximately 430 earthquakes with M ≥ 5, including 36 M ≥ 6 earthquakes, have been recorded in the region. This study aims to link the ocean bottom deformations manifested through ocean bottom pressure variations with the earthquakes' time series. To this end, we investigated the time series of the ocean bottom pressure (OBP) anomalies derived from the Gravity Recovery and Climate Experiment (GRACE) and GRACE Follow-On (GRACE-FO) satellite missions. The OBP time series comprises a decreasing trend in addition to 1.02, 1.52, 4.27, and 10.66-year periodic components, which can be explained by atmosphere, oceans, and hydrosphere (AOH) processes, the Earth's pole movement, solar activity, and core–mantle coupling. It can be inferred from the results that the OBP anomalies time series/mass change is linked to a rising trend and periods in the earthquakes' energy time series. Based on this preliminary work, ocean-bottom pressure variation appears to be a promising lead for further research.

**Keywords:** GRACE; ocean bottom pressure; earthquakes; Mediterranean Ridge accretionary complex; FFT; earthquake energy time series

## 1. Introduction

The satellite gravimetry technique has been developed to measure gravity anomalies across the Earth. The gravity anomalies are caused by mass variations, including those in ocean basins. The ocean basin mass variations are linked to changes in the hydrostatic pressure at the seafloor [1]. This hydrostatic pressure at the seafloor is known as the ocean bottom pressure (OBP). The OBP is the combined pressure caused by the column of seawater's weight and the atmosphere above the seafloor [2]. Hence, the OBP can be derived from gravity anomaly data [3].

The OBP variations are driven by (i) air mass pressure; (ii) changes in ocean water mass due to an inflow of water from the continents, and regional redistribution due to attraction effects of external masses located on the continents and in the atmosphere; and (iii) the redistribution of water within the ocean basins in response to atmospheric surface winds, atmospheric surface pressure gradients, and ocean thermohaline effects (i.e., the general ocean circulation) [1,4–9].

Plate tectonics is a distinctive source of mass change in the Earth's lithosphere. In particular, divergent and convergent plate boundaries are the places where continuous mass changes occur. A divergent boundary is an area where two plates are moving apart, and a new crust is created by magma pushing up from the mantle. A convergent boundary is an area where the plates move towards each other. When two plates meet each other, the thinner, denser, and more flexible one subducts under the other. As a result, a mass

change (horizontal/vertical movements) occurs around the subduction boundary [10–14]. Briefly, mass change in the divergent and convergent plate boundaries is reflected in OBP variations.

In this study, we investigated the link between earthquake activities and OBP anomalies. The area of interest (AOI) is the Mediterranean Ridge (MAC) accretionary complex, one of the most critical ocean bottom subduction zones globally, covering an area 300 km wide and 2000 km long. We used the OBP data from the database of the GRACE and GRACE-FO satellite missions.

## 2. Materials and Methods

The Mediterranean Ridge accretionary complex is an arc-shaped wedge located on the Eastern part of the Mediterranean seafloor. The MAC is formed by the African plate's collision with the Eurasian plate [15–17]. The MAC consists of three parts: the outer and the inner parts, and Hellenic trenches. Figure 1 illustrates the location of the MAC.

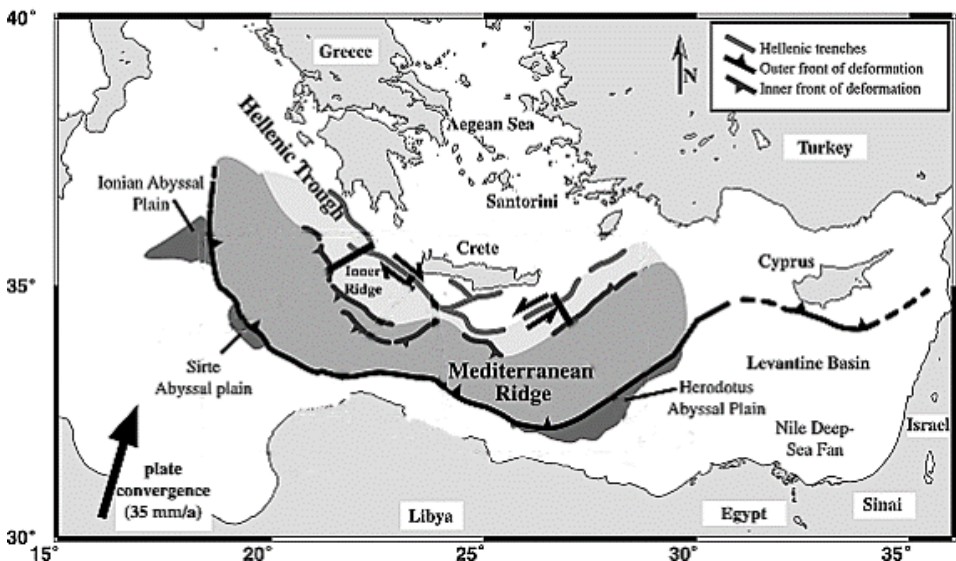

**Figure 1.** Location of the Mediterranean Ridge accretionary complex (MAC) in the Eastern Mediterranean Sea basin. The arrows indicate the present plate kinematic direction and rate between Africa and Eurasia [18,19].

The African plate in MAC subducts beneath the Eastern European plate with a relative velocity of approximately 3.5 cm/year [20–25]. The subduction zone of the MAC experiences a systematic mass change. This ridge system frequently produces major earthquakes, sometimes associated with tsunami events [26,27]. Figure 2 shows the earthquakes > M6 that have occurred in MAC since 1970.

OBP estimates were obtained from the GRACE and GRACE-FO mission. The data can be downloaded from [3]. The spatial resolution was $1° \times 1°$, and the temporal resolution was one month, except for some missing records. Consequently, on average, seven records/year were available in the dataset and the data gaps were randomly scattered throughout each year. The missing months' data were interpolated to get a uniformly spaced time series. The data cover the period from 4 April 2002 to 25 October 2017 (GRACE mission), and from 1 June 2018 to 1 July 2020 (GRACE-FO mission), which results in a seven-month-long gap in the OBP time series. The OBP data available at the source was altered by removing the time-average OBP value calculated from the period between January 2003 and December 2007 [3].

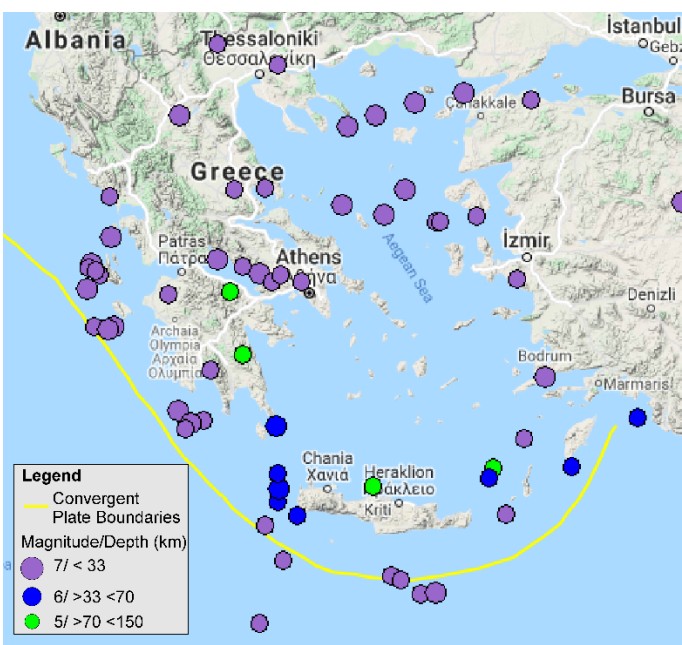

**Figure 2.** Earthquakes that have occurred since 1970 in the MedRidge zone. Source: [28].

Figure 3 shows a map of a section of the Mediterranean Sea, including the MAC region. The locations of over 200 earthquakes (M ≥ 5) are marked with black crosses. The average of all available monthly OBP data is shown as the background of the map. A spatial association between the OBP anomalies and the location of earthquakes is apparent in the MAC region.

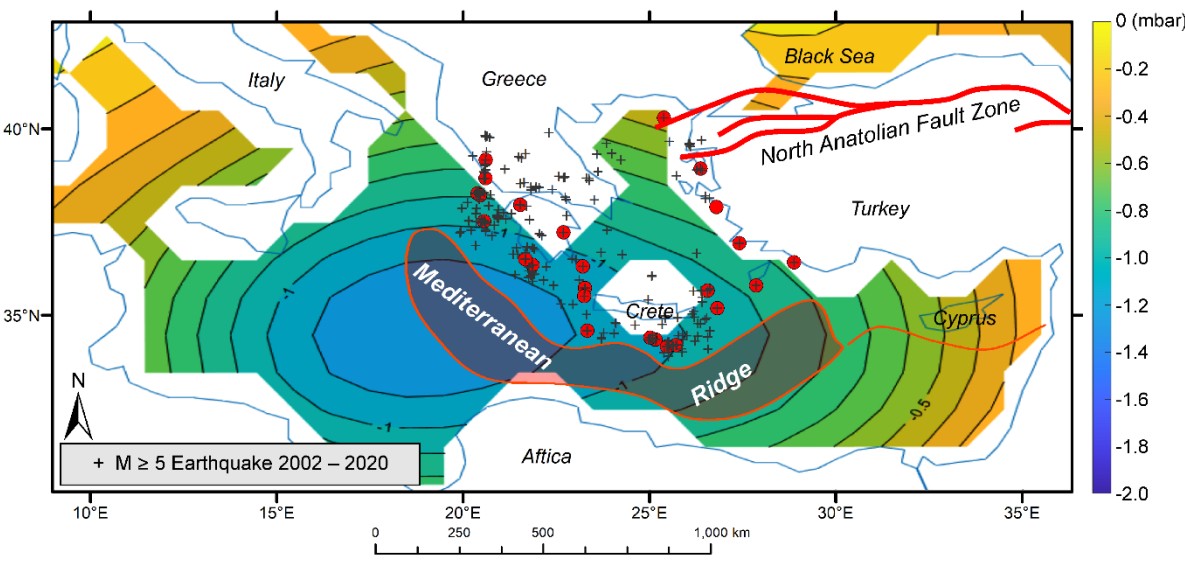

**Figure 3.** Map of the time-average ocean bottom pressure (OBP) in the MAC region and the location of earthquakes (M ≥ 5) that occurred from 2002 to the present. The red dots are M ≥ 6 earthquake locations selected for detailed study. The North Anatolian Fault zone location is marked with the red line in the map's northeastern section.

For a detailed investigation of the OBP anomalies, we selected 14 locations (M ≥ 6) where earthquakes have occurred recently in the MAC. They are shown in Figure 3 as red dots. The time series of the OBP anomalies for the 14 locations are shown in Figure 4. The grey area indicates the range of OBP for the earthquake locations. The OBP anomalies

are consistent, except for one location. The outstanding location belongs to the North Anatolian fault zone (NAFZ) and not MAC. The location of the NAFZ is shown in Figure 3.

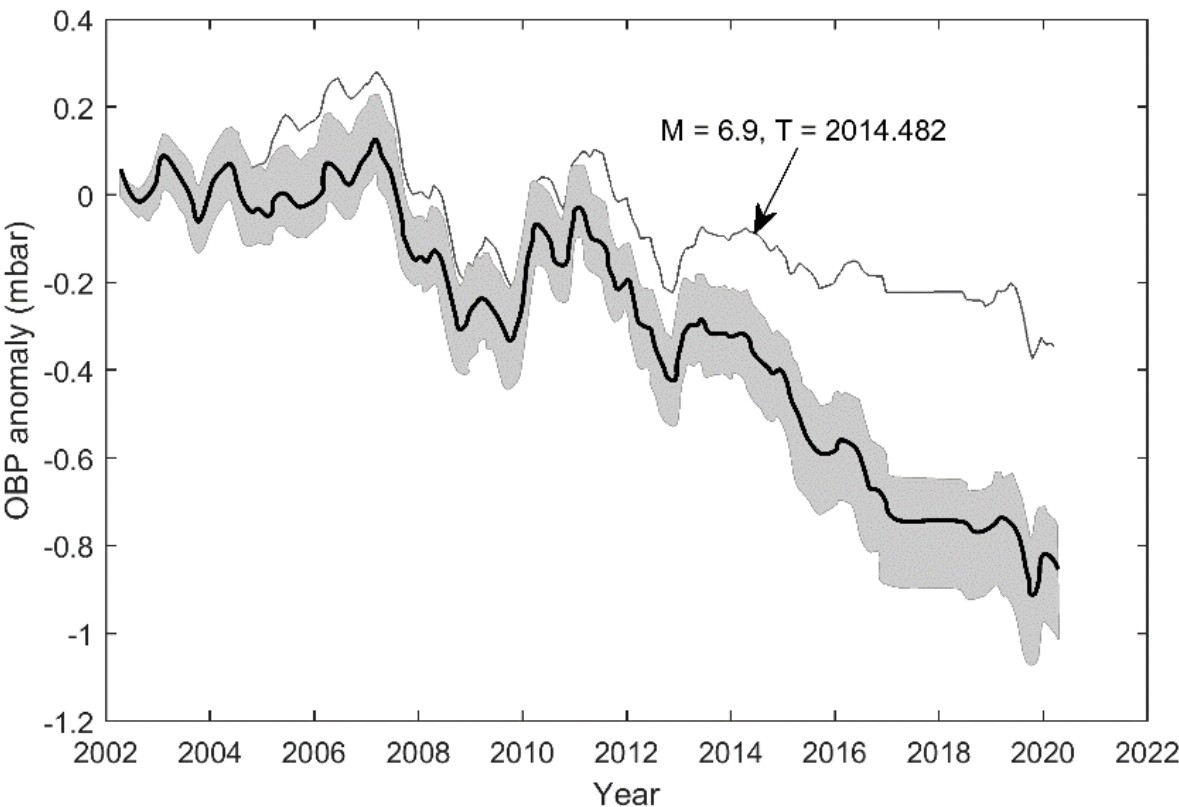

**Figure 4.** OBP anomaly range for 14 selected earthquake locations (grey area). A solid black line indicates the time-averaged OBP anomaly time series for the selected locations. The outstanding plot represents the OBP anomaly data for the M = 6.9 earthquake location (40.29N, 25.39E) at the 2014.482 epoch. This location belongs to the North Anatolian fault zone (NAFZ).

## 3. Results

The OBP anomalies time series shows a decreasing trend and oscillations. To model the OBP anomalies, we first resampled the OBP time series to a uniform sampling rate of 12 samples/yr. We detrended the time series using MatLab's piecewise linear function [29]. January 2003 and December 2017 were selected as breakpoints. This is because records between January 2003 and December 2007 were time-averaged, and the average OBP value was extracted from the data [3]. Figure 5a shows the OBP anomalies time series, the trend, and the resulting detrended OBP time series. Next, we used a third-order low-pass Butterworth filter, found in MatLab [29], to cut off frequencies higher than 0.1 cycles/month, which corresponds to 0.83-year/cycle. Using the periodogram function found in MatLab, we identified 10.66, 4.27, 1.52, and 1.02-year periods in the detrended OBP time series. Figure 5b shows the power spectral density of the OBP time series.

Here, we attempted to link the OBP anomalies trend with the tectonic activities in the MAC zone. To this end, we investigated a time series of the earthquakes' energy recorded in the MAC zone. We used Equation (1) to convert earthquake magnitude into energy [30]:

$$logE = 5.24 + 1.44M, \qquad (1)$$

where *E* is in (J).

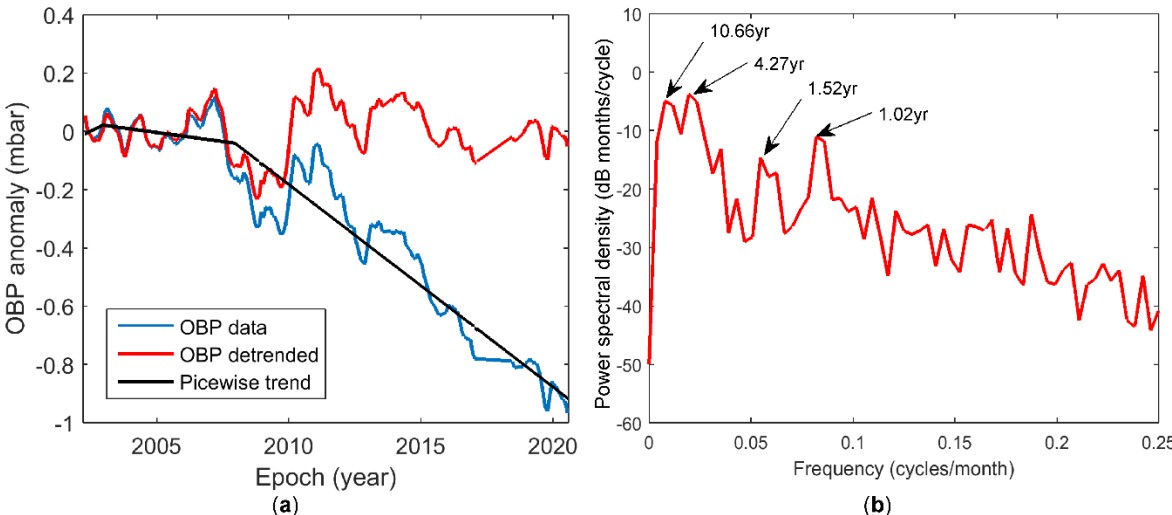

**Figure 5.** (**a**) OBP time series (blue line), piecewise trend (black line), and detrended OBP time series (red line). (**b**) Power spectrum density of the detrended OBP time series.

We summarized the energy in monthly bins. Figure 6a shows the time series of the earthquakes' energy from January 2002 to December 2020 (228 data months). A clear rising trend in the energy of earthquakes is visible. Local extremes are present in 2008, 2014, and 2020, suggesting a certain periodicity.

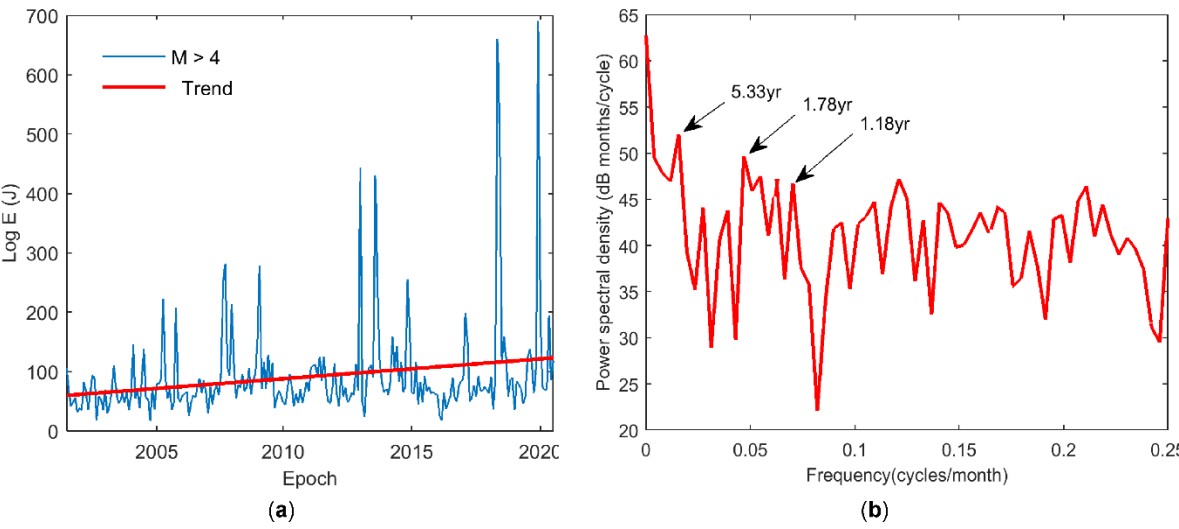

**Figure 6.** (**a**) Monthly time series of the earthquakes' energy, including the linear trend. Periodic components appear to be present as well. (**b**) The earthquake time series periodogram contains periodic components of 5.33, 1.78, and 1.18 years.

## 4. Discussion

A close study of Figure 3 shows that a significant OBP anomaly is present in the western section of the MAC zone. Such a prominent feature is not observed outside of the MAC region. This part of the ocean floor subducts under the continental part in the eastern section of the MAC zone. Therefore, a long-term decrease in the OBP anomaly in the region is anticipated because of the constant mass change.

The time-series analysis yields a prominent decreasing trend and 1.02, 1.52, 4.27, and 10.66-year oscillations in the OBP anomalies. The AOH processes play a dominant role in the Earth's dynamic processes and influence seasonal and interannual periods up to 4 years [30]. In this respect, the 1.02, 1.52, and 4.27-year oscillations are thought to be

caused by AOH processes. Longer oscillations and a linear trend in the Earth's dynamic processes are believed to be caused by a combination of geophysical causes such as the slow rebound of the crust and upper mantle [31–33] and decadal angular momentum exchange between the core and mantle [34–39]. Based on this information and the MAC's geological settings, the linear decreasing trend and 10.66-year oscillation of the OBP can be explained by the subduction tectonic process in the MAC zone, but they might also be related to the 11-year solar activity cycle.

To investigate the hypothesis that tectonic interactions in the MAC zone drive the long-term OBP variations, we compiled a time series of the annual released energy of earthquakes with M ≥ 4 for the AOI. The earthquakes' power time series features a rising trend and a periodic component. The increasing trend in the earthquake energy is consistent with the decreasing trend in the OBP and supports the hypothesis that the tectonic process mainly drives the long-term OBP variation in the MAC zone. The earthquakes' energy time series includes three oscillations with 5.33, 1.78, and 1.18-year periods. Tectonic processes drive the 5.33-year oscillation [31]. The periods of 1.78 and 1.18 years are similar to those in the OBP and are probably driven by AOH forces.

## 5. Conclusions

MedRidge is one of the most critical subduction zones in the world. This fault system has produced earthquakes > M8 that devastated the ancient cities in the region. However, such an earthquake has not been experienced for centuries. The OBP time series from GRACE and GRACE-FO satellite platforms show that the OBP around the MAC zone has gone down drastically. According to the literature, mass redistribution in the mantle and crust are among the possible reasons for OBP variation. The linear trend and 10.66-year oscillation period in the OBP data point to the effect of the subduction tectonic process in the MAC zone. The increasing linear trend in the earthquake energy released also supports this hypothesis. The MAC has the potential to produce devastating earthquakes. Therefore, the OBP variation in the region must be studied with long-term records to better understand the relationship between OBP and tectonic processes in the zone.

**Author Contributions:** H.S.K. conceived the experiment, H.S.K. and K.B. conducted the experiment, and H.S.K. and K.B. analyzed the results. H.S.K. wrote the original version. K.B. edited the text. All authors have read and agreed to the published version of the manuscript.

**Funding:** This research received no external funding.

**Institutional Review Board Statement:** Not applicable.

**Informed Consent Statement:** Not applicable.

**Data Availability Statement:** No new data were created or analyzed in this study. Data sharing is not applicable to this article.

**Acknowledgments:** The authors thank the Estimating the Circulation and Climate of the Ocean (ECCO) Project Consortium and JPL NASA for the OBP data.

**Conflicts of Interest:** The authors declare no conflict of interest.

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
