# Peer review of "Analysis of Ocean Bottom Pressure Anomalies and Seismic Activities in the MedRidge Zone"

_remotesensing, doi:10.3390/rs13071242_

Round 1

Reviewer 1 Report

Dear authors,

thanks for this study. It is well written but the scientific content requires significant improvements. The simple FFT-approach you apply to the data is not sufficient to extract the conclusions you draw. Basically you detect numerical artifacts which are related to the total duration of the time series, i.e. 18 years divided by 2, 3, ...

Please check the attached PDF for further comments, questions  and suggestions.

Kind regards

Reviewer

Author Response

Thank you for your excellent work.

Best Regards,

Authors

Reviewer 2 Report

Basic comments and objections:

- Figure 3 shows the locations of earthquakes M> = 5. Has an attempt been made in calculations to use earthquakes e.g., M> = 4 or does it make no sense?

- At the beginning of Chapter 3, the application of the piecewise linear function found in MatLab is mentioned. Can you write a little more about it, regardless of what the Letter article is about here?

- In the continuation of the same chapter it is said that to filter out high-frequency data, we used the low-pass Butterworth filter. It is necessary to say a little more about the filter and certainly to cite the literature.

- It would be good to cite the literature for the formula (1) used.

- Figure 1 shows whether the lines are perpendicular to the MAC extension (3, 6, 18/19, 30). Do they have a role in the article or is it just a matter of taking a picture from the literature?

- The Discussion chapter states that the increasing trend of the earthquake energy is consistent with the decreasing trend of the OBP. This is clearly seen in Figures 5a and 6a as well. Therefore, the question arises: would it make sense to look for a correlation coefficient between them?

Author Response

Thank you for your excellent work. We really appreciate your contribution.

Best Regards,

Authors

Reviewer 3 Report

Please see my comments in the attached PDF.

Author Response

(The authors gave the same response as above.)

Round 2

Reviewer 3 Report

I do not have further comments.

Author Response

Thank you for your time to review our manuscript.

Best regards,

Authors